# A Novel Boot Camp Program to Help Guide Personalized Exercise in People with Parkinson Disease

**DOI:** 10.3390/jpm11090938

**Published:** 2021-09-20

**Authors:** Josefa Domingos, John Dean, Travis M. Cruickshank, Katarzyna Śmiłowska, Júlio Belo Fernandes, Catarina Godinho

**Affiliations:** 1Department of Neurology, Center of Expertise for Parkinson & Movement Disorders, Donders Institute for Brain, Cognition and Behaviour, Radboud University Medical Center, 6500HB Nijmegen, The Netherlands; domingosjosefa@gmail.com; 2Triad Solutions, Aurora, CO 80012, USA; john@johnmdean.com; 3Grupo de Patologia Médica, Nutrição e Exercício Clínico (PaMNEC)—Centro de Investigação Interdisciplinar Egas Moniz (CiiEM), Escola Superior de Saúde Egas Moniz, Caparica, 2829-511 Almada, Portugal; juliobelo01@gmail.com; 4Centre for Precision Health, Edith Cowan University, Joondalup, WA 6027, Australia; t.cruickshank@ecu.edu.au; 5Perron Institute for Neurological and Translational Science, Nedlands, WA 6009, Australia; 6Department of Neurology, Silesian Center of Neurology, 40-055 Katowice, Poland; kasia.smilowska@gmail.com

**Keywords:** Parkinson disease, personalized medicine, clinical exercise, exercise prescription, boot camp, physiotherapy

## Abstract

Given the variety of exercise programs available for people with Parkinson’s disease (PD), such individuals may struggle to make decisions about what exercise to perform. The objective of this study was to assess the usefulness, satisfaction, and preferences regarding participation in a PD-personalized educational and exercise boot camp program. Attendees participated in a four-day program consisting of exercise sessions, workshops, and social activities. We collected demographic and clinical information. We assessed satisfaction and preferences immediately after. At one-month follow-up, participants assessed usefulness and changes in exercise habits. Eight individuals diagnosed with PD, with a mean age of 59.5 ± 6.8 years, participated. All participants felt “very satisfied” and likely to attend future events. The two favorite sessions were: cognitive stepping and dance-based movements. At one-month follow-up, participants considered the program “very useful” and reported changes in their exercise routine. Our results suggest that the boot camp program was considered useful and capable of influencing participants’ exercise habits.

## 1. Introduction

Parkinson’s disease (PD) is a neurodegenerative disease that results in a gradual reduction in activities of daily living and quality of life [1]. Increasing evidence suggests that individuals with PD benefit from continuous ongoing exercise to improve and maintain physical functioning and help them better manage the disease [2,3].

As such, people with PD are currently encouraged to play an active role in self-management and acquire the exercise tools in their community to manage their disease [4]. Yet, there is a variety of rehabilitation, exercise, and physical activity programs that are now available for individuals with PD. Some of the more common approaches highlighted in recent literature include amplitude-based movements [5,6,7,8], dance [9,10,11], Tai Chi [12,13], Qigong [14,15], Nordic walking [16,17], boxing [18,19,20], and aquatic exercise [21,22].

Translation of therapies into practice in the community setting has, however, proven difficult. Engaging in a regular rehabilitation and clinical exercise interventions and guaranteeing ongoing adherence requires finding the proper program that really fits the patients’ needs and preferences, as well as bypasses common perceived barriers to exercise in PD [23,24].

Just telling people about the benefits of these exercises and expecting them to make decisions is debatable. Making the choice about which type of exercise programs to perform, understanding the evidence, safety issues, and the adaptations needed for each program [25] may be difficult. Considering the benefits of such programs, there is a need to overcome these translation issues.

Several lines of evidence have suggested that boot camps represent an effective and useful means of teaching [26,27]. Despite knowledge of this success, few studies have investigated the usefulness of delivering exercise education via boot camps for individuals with PD.

No study has attempted to encompass the range of these different approaches all in one intervention or program in PD. Providing access to a boot camp program that provides participants the opportunity to familiarize themselves with types of exercise programs while guiding them as to what works best and potentially guiding their exercise choices may be of critical importance. Here, we present a PD-personalized educational and physical activity boot camp program that provides participants the opportunity to familiarize themselves with common evidenced exercise programs.

Our primary objective was to assess the patients’ perceived usefulness (or helpfulness) and utilization (or how much they changed their exercise routines and made use of the knowledge) of the program to facilitate exercise choice and lifestyle behavior over time. Secondary objectives included the assessment of satisfaction, preferences, and adverse events.

## 2. Methods

### 2.1. Design

We used a single group pre-test and post-test design.

### 2.2. Sampling and Recruitment

The sampling method selection was non-probabilistic by convenience. Following European Parkinson guidelines regarding the recommended number of individuals per group [3], we included eight individuals with PD, invited via email from a Parkinson’s patient association. All participants had to be able to ambulate independently, able to tolerate a minimum 1 h of exercise, and able to attend all 4 days of the boot camp. Care partners were also invited to attend.

### 2.3. Participants

Eight individuals with PD (5 women) participated in the program with a mean age of 59.5 ± 6.8 years. All participants were participating for the first time in a boot camp. The participants’ demographics and clinical characteristics can be found on Table 1.

### 2.4. Ethics and Procedures

This study follows the principles of the Declaration of Helsinki. The study was approved by the Egas Moniz Research Ethics Board. Participants completed an informed consent form before starting the program and received an information sheet explaining how their data would be used.

### 2.5. Program

The boot camp program consisted of four days of training featuring sessions that were 30–60 min each for a total average of 4 h per day. The boot camp began with an overview of general group goals based on a pre-assessment questionnaire. After, it focused on alternating between exercise, educational sessions, and social patient–family interactions. See Table 2 for topics presented on each day. To provide an engaging educational experience, therapists used various formats to teach, including multimedia presentations, hands-on activities during lectures, exercise sessions, and interactive discussions (question-and-answer periods that allowed participants to learn from the therapists but also from each other). The tips and tricks sessions, for example, included trying the cueing strategies taught and exploring solutions for better management of issues associated with PD reported on a pre camp questionnaires. Presenters’ slides and tangible information to easily reference later were shared with all participants.

The exercise sessions included practicing examples of amplitude-based interventions [5,6,7,8], Nordic walking [16,17], multitask cognitive and motor exercise challenges [2,28,29,30], hydrotherapy [21,22,31] and tai-chi [12,13], dance-based movements [9,10,11], as well as a boxing session [18,19,20].

All sessions were delivered in a group format. To guarantee safety, two volunteers were available to provide support to people if needed. The intensity of the physical activity sessions was delivered according to patient’s tolerance and started ranging from a low to moderate intensity of aerobic exercise (40–60% HRR (or VO2R). Due to the group format, patients were taught to self-assess and monitor their effort throughout each session. The volumes of the aerobic based sessions were ≥30/40 min of continuous or intermittent exercise per session based on the exercise recommendations in European guideline for physiotherapy for Parkinson’s disease [3]. The program was led by a physiotherapist and a speech therapist with specialized training and experience in PD (Josefa Domingos and John Dean). A local physiotherapist was invited to the program to be able to carry on exercise classes if requested by participants.

Social activities (e.g., mealtimes, resting breaks) promoted interaction with peers and contributed to peoples’ feeling more acquainted with each other, adding to a positive group dynamic.

### 2.6. Data Collection

Participants completed an online pre-assessment structured questionnaire before the program, a post-assessment immediately after, and a one-month follow-up.

The pre-assessment collected general information on demographics, top 5 clinical problems, past medical conditions, disease management strategies used, current exercise, preferred exercise modalities, and perceived barriers/facilitators to performing exercise. The pre-assessments also inquired about the participant’s goals and expectations for attending the boot camp.

Immediately after the program, we assessed satisfaction, preferences, and any adverse events via online anonymous questionnaire. Patients were asked to rate on 4-point scales how satisfied they were with boot camp (1 = not satisfied, 2 = neither, 3 = satisfied, and 4 = very satisfied), whether they would recommend the program to a friend (1 = Yes, 2 = No, 3 = Maybe), and how likely are they to return to a similar program (1 = very likely, 2 = likely, 3 = neither, and 4 = unlikely). They were also asked which sessions they preferred (1 = Educational sessions; 2 = Dance-based amplitude movements; 3 = Voice & Breathing; 4 = Boxing session; 5 = Walk & Talk dual task training; 6 = Cognitive stepping program; 7 = Hydrotherapy activity; 8 = Nordic walking; 9 = On the floor activities; 10 = Drum-based dance activity) and if there was any type of problem encountered during the program.

A follow-up anonymous questionnaire emailed to each participant one month after the boot camp asked the participants to assess the perceived usefulness of the program in the management of the current participant’s exercise habits, as well as to assess the utilization (or how much they changed their exercise routines and made use of the knowledge and experience of the bootcamp). Usefulness was assessed based upon the patients’ perceived (self-reported) usefulness of the program. Patients were asked to rate on 4-point scales how useful the boot camp was in helping them manage their current exercise habits (1 = not useful, 2 = neither useful, nor useless, 3 = useful, and 4 = very useful). Utilization was based upon changes in exercise habits that were made after the program.

### 2.7. Data Analysis

Data were extracted to a spreadsheet. Using the IBM Statistic Package for the Social Sciences software, version 26.0 descriptive statistics were adopted to analyze data.

## 3. Results

Participants most frequently reported their expectation for the boot camp was to learn more about specialized exercise. See patients’ exercise habits and expectations for the boot camp in Table 3.

### 3.1. Percieved Usefulness and Utilization

Participants considered the boot camp very useful (seven in eight; 87.5 %) and useful (one in eight; 12.5%) in managing their current exercise habits. At one-month follow-up, seven participants reported making changes in their exercise routine after the boot camp, and one participant said, “Not really changed anything; I need more time in my day”. The participants that made changes mentioned specifically: “I do exercise more frequently now”; “Introduced the “power” breathing that I learnt in the program”, “Taking daily commitment to exercise seriously”, “Joined a gym to do more exercise”, “I do more walking periods and cognitive games”, “I started Nordic walking” and, “started sessions with the physiotherapist involved in the program”.

### 3.2. Satisfaction

After the program, all participants had favorable feedback, with all (100%) feeling very satisfied, likely to attend future events, and “would recommend to another person with PD”.

### 3.3. Preferences

About six in eight (75 %) participants felt that many of the exercises and activities were new to them. The two training sessions that participants enjoyed the most were: dual task cognitive stepping (six in eight; 75 %) and dance-like movements (six in eight; 75 %). The two sessions that participants enjoyed less were: Nordic walking (two in eight) and boxing training (two in eight).

### 3.4. Safety

No falls, major injuries, or adverse events occurred during the program. One patient could not attend boxing sessions due to a superficial finger injury.

## 4. Discussion

Our study aimed to assess the usefulness, utilization, satisfaction, and preferences regarding participation in a PD-personalized educational and clinical exercise boot camp program. Based on our results, the boot camp was perceived to be useful by all eight participants with mild and moderate PD. Overall, seven out of eight participants referred they had made changes in their exercise routines one month after. The patients reported high satisfaction levels and there were no severe adverse events. Given the benefits of ongoing exercise in PD and the number of current options, the importance of offering guidance for safe, PD-specific, exercise programs for this population cannot be underestimated [18,25]. Importantly, this paper could help pave the way forward to delivering personalized exercise education through online boot camps during the COVID-19 pandemic crisis.

After the program, the patients were highly satisfied. Several positive aspects may have contributed to the high patient satisfaction reported. First, the information from the questionnaire prior to the program allowed for better tailoring of the program to all the participants’ needs. Besides clinical information on health status, the questionnaires also provided with logistical information based on questions such as “What would be a good time for you to start the activities?”, “If you are bringing someone, please tell us who and what would you like them to learn from the boot camp?”, and “What topics and activities would you like to be addressed in the boot camp?”. Personalized exercise programs that are appropriately designed and delivered increase patient satisfaction with care options [32]. Shared goal setting should always be considered during the initiation and implementation of exercise interventions. Second, intrinsic motivation, via experiencing the actual benefit of each exercise, and enjoying participation are important factors for long-term training adherence [33,34].

The ability to try out the exercises can make a significant difference in one’s ability to judge the preference, usefulness, and long-term utilization of the varied exercise options. It ultimately determines the degree to which individuals can judge information effectively and use it to make smart exercise selections. Third, delivering the program in a group format allowed participants to experience the beneficial effects of social support and motivation from fellow patients [35]. Fourth, most of the participants (six in eight; 75 %) felt that many of the exercises and activities were new to them, even though all participants were already doing some sort of exercise on a weekly basis. They reported initially having clear expectations for the boot camp to gain more knowledge on specific PD exercises. Accessing specialized care was considered initially as one of the most common factors that would facilitate exercising (six in eight participants).

Implementing such a program was not without some limitations. First, some patients might not have future access to personalized exercise programs presented. As such, we invited a local physiotherapist to the program to help ensure that participants would have an opportunity to access specialized care if needed. This was an innovated solution to facilitate access to personalized care. Interestingly, most participants preferred to include changes in their daily routine activities with only one preferring to book sessions with the physiotherapist after completing the program. However, measuring the knowledge of participants immediately after the program would have brought better insights concerning reasons for change or no change (utilization) one month after. Second, even though participants were told to invite their families and caregivers, only one participant decided to bring a caregiver. Nevertheless, all participants were confident that they could contact the physiotherapist any time they needed additional information or insight. The influence of the caregiver should be a key factor to be better explored in future programs. Third, the group format imposed some safety concerns. We limited registrations to eight people with PD to assure a safe and efficient setting according to current recommendations [3]. Fourth, the nature of the study, with a small sample size and single center design, imposes restrictions on the generalizability of the findings. We included a heterogeneous group of people with PD with different backgrounds and did not specifically include participants with more advanced stages of PD. Further research is needed to measure its effectiveness on other subtypes and profiles of people and against other educational programs.

## 5. Conclusions

Our findings demonstrate that trying out different exercises can support judgments and decisions and thus may be a viable tool for more effective exercise prescription programs for the PD community. It is easily replicable, improves patients’ knowledge about exercise, and based on the reported interest of participants in attending another similar program as well as the benefits they perceived of the usefulness of the program, is likely to be well-received by health care professionals world-wide.

## Figures and Tables

**Table 1 jpm-11-00938-t001:** Participants’ general and clinical characteristics.

Participant	Gender	Age	Time since Diagnose	Main Problems	Perceived Health (Now and Compared to Last Year)	Fall History
1	F	66	10 years PD	My inability to help my husband in a casual spontaneous way The lack of reliability and independence Fear of loss of mind, dignity, feeling powerlessness Fear of falling Dyskinesia and appearing to be drunk	Poor, the same as last year	1 fall outdoors, 2 months ago Near falls occasionally
2	F	61	2.8 years PD	Slowness Balance and walking Speech & swallowing Apathy	Very good and about the same	1 fall outdoors 6 months ago, couldn’t get up. Near falls occasionally
3	M	59	8 years PD	Stiff neck and bad posture Walking	Good, about the same	No falls Near falls occasionally
4	M	60	8 years PD	Walking indoors Walking outdoors Time management & relaxing	Good, somewhat worse than one year ago	Frequent falls. Weekly, don’t pick my feet up. Daily near falls A little fear of falling
5	F	57	4 years PD	Walking Walking and talking	Good, somewhat worse than one year ago	No Falls No Near falls
6	F	70	6.5 years PD	Imbalance Walking coordination Anxiety, frustration, self-confidence Driving, being able to see in the dark	Fair, much less now than a year ago	2 Falls last 6 months Near falls once per week
7	F	60	1.8 years PD	Night (early morning) and morning stiffness (pre-medication) OFF phases Lower back and sometimes neck pain Memory/ Concentration—recalling names or a particular word Being able to stay focused on one thing	Fair, much less now than a year ago	No Falls No near falls Some fear of falling
8	M	45	9 years PD	Lack of sleep, Restlessness Freezing/walking Unfit/overweight Stiffness	Good, about the same	No Falls Near falls occasionally

**Table 2 jpm-11-00938-t002:** Description of the boot camp educational and exercise sessions’ topics.

Days	Educational and Exercise Sessions	Format
Day 1	One-on-one brief assessments and introductions	Education
	Exercise in Parkinson Disease: why, what and how.	Education
	Introduction to variety of exercises.	Practice
	New exercise ingredient in PD: cognitive-motor training.	Education
	Cognitive Stepping with amplitude-based movements.	Practice
Day 2	Review of Day 1, goals for the day	Education
	Changes in voice, communication and swallowing in PD	Education
	Voice, Breathing & Movement session	Practice
	Educational session: Walking/freezing/talking difficulties	Education
	Walk & Talk dual task program	Practice
	Nordic walking with integrated voice training session	Both
Day 3	Review of Day 2, goals for the day	Education
	Hydrotherapy: impact on mobility	Education
	Exercise Session: Using boxing for mobility and voice training in PD	Practice
	Tips & Tricks for transfers in daily life and exercise	Both
	Exercises for transfers: rhythm, amplitude & speed	Practice
Day 4	Review of the Camp, future goals & community resources	Education
	Tips & Tricks for bypassing barriers to exercise.	Education
	Goodbye Drum-based dancing activity	Practice
	Adapted Tai Chi	Practice
	‘Take home messages’ test activity	Education

**Table 3 jpm-11-00938-t003:** Participants’ current exercise habits, perceived exercise barrier’s, facilitators and expectations for the boot camp.

Participant	Current Exercise Habits	Perceived Barriers to Exercise	Perceived Factors That Would Facilitate Exercising	Expectations for the Exercise Camp Program
1	Current: yes Which: stretching Frequency: daily Average time: 15 min	Fluctuations in health	Caregiver/partner are supportive, accessing instructors with PD expertise	“To try to find a way back, relearn how to exercise, get a shared goal. Learn from others. Reduce need to take pills”
2	Current: yes Which: walking, ball activities. Frequency: every other day Average time: 20 min	Poor coordination Fear of looking silly Accessing instructors with PD expertise is difficult	Perceived benefit and visible improvement after exercising, able to work out during working hours	“Improve my coordination and health without the use of drugs”
3	Current: yes Which: cycling, walking Frequency: cycling twice a week, walking every day. Average time: 60 min	Fluctuations in health Not easy to access specialized exercises	Perceiving benefit and visible improvement after exercising, accessing instructors with PD expertise	“To understand what exercises would be best for me, also being with fellow sufferers of similar abilities”
4	Current: yes Which: Cycling Frequency: 1 per week Average time: 45 min	Fluctuations in health Lack of time to exercise Always something to do for Parkinson society	Perceiving the benefit after exercising	“To access the “feel good factor” & challenge myself more, see what I should do for exercise”
5	Current: yes Which: walking the dog Frequency: Daily Average time: 30 min	Fluctuations in health Lack of time to exercise Job restrictions	Accessing instructors with PD expertise	“I would like to gain knowledge and further my skills in managing the disease through exercise”
6	Current: yes Which: golf, walking Frequency: 2 times a week Average time: 60 min	Fluctuations in health Lack of confidence exercising outside of rehab setting Weather conditions Living in a rural area without access to specialized care	Perceiving benefit and improvement after exercising, easy access to the gym or exercise facility, accessing instructors with PD expertise, fun things in group with people fitness same as me	“To boost confidence and fitness levels, in fun environment”
7	Current: yes Which: Parkinson Choir, Conductive Education Frequency: choir 1 per week; education 1 per month Average time: 75 min	Fluctuations in health Fear of falling, effect of weather conditions	Caregiver/partner being supportive, easy access to the gym or specific exercise facility, accessing instructors with PD expertise	“To identify a daily/weekly exercise regime with ones that will be more beneficial to me”
8	Current: yes Which: Jiu-jitsu Frequency: once a week Average time: 60 min	Fluctuations in health Lack of time to exercise Lack of interest Job restrictions	Easy accessing to the gym or exercise facility, accessing instructors with PD expertise, able to workout during working hours	“Get tips for getting fitter and do more specific exercises”

## Data Availability

The data presented in this study are available on request from the first author.

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
