# Peer review of "A Novel Boot Camp Program to Help Guide Personalized Exercise in People with Parkinson Disease"

_jpm, 2021, doi:10.3390/jpm11090938_

Round 1

Reviewer 1 Report

Even though the paper is well written and the quality of tables is high,  it needs a little more improvement. First of all, authors did not describe exercise volume and intensity. Moreover,  exercises usefulness  was evaluated only on the basis of subjective parameters and this limits the scientific value of the paper. 

Author Response

We thank the reviewer for these additional suggestions. In light of these suggestion, we highlight our modifications and responses below.

Related to the description of exercise volume and intensity, we have added this within the text:

Lines 108 to 114: Intensity of the physical activity sessions was delivered according to patient’s tolerance and started ranging from low to moderate intensity of aerobic exercise (40% - 60% HRR (or VO2R). Due to group format patients were taught to self-assess and monitor their effort throughout each session. The volume of the aerobic based sessions were ≥30/40 min of continuous or intermittent exercise per session, based on the exercise recommendations in European guideline for physiotherapy for Parkinson’s disease [3].

Related to the comment: “Moreover, exercises usefulness was evaluated only on the basis of subjective parameters and this limits the scientific value of the paper.”

The primary goal of this research was to contribute to fill a need for Parkinson specific knowledge and experience with exercises; to offer something people with Parkinson’s would find useful. We focused on qualitive data. We have better clarified terminology used to better reflect our intentions and added “…to assess the patients’ perceived usefulness (or helpfulness) and utilization (or how much they changed their exercise routines and made use of the knowledge)”. Lines 65 to 67.

We added in the outcome of utilization as a means to better represent the assessment of how much people will use the information and knowledge provided in the program. This reflects that they used the learnings and experienced of the program that aimed at promoting changes in their lifestyle. We defined usefulness as the patient perceived usefulness (even if they did not change behaviors). We thus assess a) patient perception of usefulness (usefulness) and b) did they change habits after 1 month (utilization).

Our findings have implications for preliminary data regarding the understanding of efficient change following such an educational treatment program. We believe longer-term evaluations of patient ratings of usefulness and utilization are needed and are important for providing a window into skills patients can use independently. We recognize that we could have benefited from measured objectively the knowledge before and after. As such, we have highlighted it in the discussion and added the following sentence (lines 222 to 224): “However, measuring the knowledge of participants immediately after the program would have brought better insight on reasons of change or not change (utilization) one month after”.

We have also changed all references of to the word “usefulness” to “patient perceived usefulness” to better clarify what we did. 

Reviewer 2 Report

This is a well writen paper. It reports on the effects in eight patients. This number is too low to allow any conclusions. Please reject it.

Author Response

We thank the reviewer for their comments. Regarding the comment: “It reports on the effects in eight patients”, we have highlight in the discussion that the reason why we limited the group to 8 people was an effort to follow the European guidelines for physiotherapy in Parkinson’s disease, which recommends that group sizes be limited to a maximum of 8 per therapist, (“We limited registrations to 8 people with PD to assure a safe and efficient setting according to current recommendations [3]). We also recognize the limitations of this number in the discussion: “… the nature of the study, small sample size and single center design, imposes restrictions on the generalizability of the findings”. Testing group interventions in Parkinson’s in a safe manner will require small amount of people to guarantee safety and efficacy of the intervention.

Additionally, this is a qualitative study with a “proof-of-concept” nature which helps us reflect upon our primary outcome and add on preliminary evidence of the program’s usefulness and utilization at 1 month. As this is the first known study to attempt to encompass a range of different approaches all in one intervention program for PD, it has its merit and we hope that the novelty of our approach should be taken into account.

Additionally, we wish to share that this type of sample size has also been reflected in other pilot studies in the Parkinson area, many of which are usually composed of small samples before larger efficacy studies may be designed. Examples include:

McKee KE, Johnson RK, Chan J, Wills AM. Implementation of high-cadence cycling for Parkinson's disease in the community setting: A pragmatic feasibility study. Brain Behav. 2021;11(4):e02053. doi:10.1002/brb3.2053. (7 participants)

Fontanesi C, DeSouza JFX. Beauty That Moves: Dance for Parkinson's Effects on Affect, Self-Efficacy, Gait Symmetry, and Dual Task Performance. Front Psychol. 2021;11:600440. Published 2021 Feb 5. doi:10.3389/fpsyg.2020.600440. (7 participants)

Combs, S.A., et al., Boxing training for patients with Parkinson disease: a case series. Phys Ther, 2011. 91(1): p. 132-42 (6 participants)

Pantelyat A, Syres C, Reichwein S, Willis A. DRUM-PD: The use of a drum circle to improve the symptoms and signs of Parkinson's disease (PD). Mov Disord Clin Pract. 2016;3(3):243-249. doi:10.1002/mdc3.12269 (10 participants)

McNeely ME, Mai MM, Duncan RP, Earhart GM. Differential Effects of Tango Versus Dance for PD in Parkinson Disease. Front Aging Neurosci. 2015;7:239. Published 2015 Dec 21. doi:10.3389/fnagi.2015.00239 (11 participants)

Round 2

Reviewer 2 Report

Please accept it